# The CRISPR-Cas9 System in *Entamoeba histolytica* Trophozoites: *ehcp112* Gene Knockout and Effects on Other Genes in the V1 Virulence Locus

**DOI:** 10.3390/microorganisms13092219

**Published:** 2025-09-22

**Authors:** Luz Virginia Reyes, Guillermina García-Rivera, Rosario Javier-Reyna, Edgar Morales-Rios, Sergio Tinajero, Cecilia Bañuelos, Daniel Talamás-Lara, Esther Orozco

**Affiliations:** 1Departamento de Infectómica y Patogénesis Molecular, Centro de Investigación y de Estudios Avanzados del Instituto Politécnico Nacional (Cinvestav-IPN), Av. IPN # 2508, San Pedro Zacatenco, GAM, Ciudad de México 07360, Mexico; luz.reyes.glez@cinvestav.mx (L.V.R.); gugarcia@cinvestav.mx (G.G.-R.); rjavier@cinvestav.mx (R.J.-R.); 2Departamento de Bioquímica, Centro de Investigación y de Estudios Avanzados del Instituto Politécnico Nacional (Cinvestav-IPN), Av. IPN # 2508, San Pedro Zacatenco, GAM, Ciudad de México 07360, Mexico; edgar.morales@cinvestav.mx (E.M.-R.); aaron.tinajero@cinvestav.mx (S.T.); 3Departemento de Investigación y Estudios Multidisciplinarios, Programa Transdisciplinario en Desarrollo Científico y Tecnológico para la Sociedad, Centro de Investigación y de Estudios Avanzados del Instituto Politécnico Nacional (Cinvestav-IPN), Av. IPN # 2508, San Pedro Zacatenco, GAM, Ciudad de México 07360, Mexico; cebanuelos@cinvestav.mx; 4Unidad de Microscopía Electrónica, Laboratorios Nacionales de Servicios Experimentales (LaNSE), Centro de Investigación y de Estudios Avanzados del Instituto Politécnico Nacional (Cinvestav-IPN), Av. IPN # 2508, San Pedro Zacatenco, GAM, Ciudad de México 07360, Mexico; dtalamas@cinvestav.mx

**Keywords:** CRISPR-Cas9 system, *Entamoeba histolytica*, *Ehcp112* gene, V1 virulence locus, Genome-editing

## Abstract

Gene editing enables a better understanding of protein functions. The genome of the protozoan parasite *Entamoeba histolytica* contains a 4500 bp DNA fragment comprising the *ehcp112*, *ehadh*, and *ehrabb* genes, which together form the V1 virulence locus. Studying these genes has been challenging due to the lack of suitable methodologies. Here, we report the first *in vitro* and *in vivo* knockout in *E. histolytica* (*ehcp112* gene) using a modified CRISPR-Cas9 strategy and explore its effects on the other V1 locus genes. Confocal and transmission electron microscopy were used to detect the RNP pathway formed by the Cas9 enzyme and the crRNA–tracrRNA complex, from their entry into the trophozoites until their arrival at the nucleus and heterochromatin. Scanning electron microscopy revealed that the mutant cells (EhCP112-KO) were smaller, with fewer pseudopodia and plasma membrane depressions. DNA sequencing and RT-qPCR assays identified a four-base deletion in the *ehcp112* gene in the mutant trophozoites. Western blot assays of EhCP112-KO extracts revealed the absence of the EhCP112 protein. While the EhCP112-KO lysates digested gelatin more efficiently than the HM1:IMSS extracts, their secreted products showed poor enzymatic activity. The *ehcp112* knockout also affected the transcription of the *ehadh* and *ehrabb* genes, probably due to their genomic position. In conclusion, the implementation of the CRISPR-Cas9 strategy in *E. histolytica* evidenced the coordinated expression of the *ehcp112* gene and the other members of the V1 locus.

## 1. Introduction

*Entamoeba histolytica* is the protozoan responsible for human amoebiasis, an endemic disease in tropical areas that causes 100,000 deaths each year. This parasitosis has also been diagnosed in non-endemic regions [1]. Infection occurs when humans ingest water or food contaminated with cysts, the infective form of the parasite. In the stomach, the cysts begin their conversion to trophozoites, the invasive form. Trophozoites invade the intestinal epithelium and can reach the liver through the portal vein, causing abscesses that, if left untreated, are lethal. In fact, trophozoites are capable of invading and destroying any human tissue [2].

Clone L-6 was the first stable phagocytosis-deficient mutant generated in *E. histolytica*. It was obtained by treating trophozoites of the virulent HM1:IMSS (HM1) strain with ethyl methane sulphonate (EMS) [3]. Using these mutants and the revertant cells, we identified the EhCPADH complex formed by the EhCP112 cysteine protease (50 kDa) and the EhADH112 adhesin (75 kDa), proteins involved in the processes of adhesion, cytolysis, and phagocytosis, which are key virulence mechanisms of the parasite. However, the EMS reagent induces pleiotropic genomic alterations and randomly affects multiple genes. Moreover, at that time, the amoeba genome had not yet been annotated, making in-depth studies and data interpretation difficult.

Gene knockdown has been performed in *E. histolytica* trophozoites after the description of the abundant 27-nucleotide (nts) RNAs that the parasite possesses in its nucleus. These 27 nts small RNAs modify histones and regulate gene editing at the transcriptional level [4]. This finding enabled the use of interference RNA (iRNA) to mutate genes [5], significantly improving the ability to genetically manipulate *E. histolytica*. Nevertheless, the lack of selection markers, its genome’s plasticity, and the presence of multicopy gene families, as well as DNA recombination events in trophozoites, represent major challenges for obtaining knockout mutants [6]. A deeper understanding is required to advance our comprehension of the genes and proteins involved in the molecular mechanisms through which this parasite invades the host. Therefore, the implementation of methodologies for generating knockout parasites is essential.

Mendes Kangussu-Marcolino et al. (2021) [7] reported the application of the powerful CRISPR-Cas (clustered regularly interspaced short palindromic repeats and Cas endonuclease) strategy in *E. histolytica*. They transfected trophozoites with a plasmid carrying the luciferase gene. Then, they used a specific guide RNA (gRNA) with the Cas9 nuclease [7] to knock out the exogenous gene. Their positive results opened the possibility of applying this strategy in *E. histolytica*.

The CRISPR-Cas9 system consists of the *Streptococcus pyogenes* Cas9 endonuclease, which specifically cleaves DNA target using a sgRNA containing a 20 nts guide sequence that directly matches the target site. The only requirement for Cas9 target site selection is the presence of an adjacent NGG motif (PAM) in the gene. Cas9 creates a double-stranded break (DSB) in the DNA. The DSB could be repaired by a random mutation or by an exogenous DNA fragment (repair template) [8,9]. The application of this powerful methodology has encountered challenges in protozoa, but the advances have also been remarkable in several parasites. In *Toxoplasma gondii*, genes previously uneditable by conventional approaches have been successfully modified using the CRISPR-Cas9 strategy [10]. This methodology has also been applied in *Plasmodium* [11], *Trypanosomatidae* [12], *Cryptosporidium* [13], *Leishmania* spp. [14], and *Trichomonas vaginalis* [15], among others. Here, we report the first *in vitro* and *in vivo E. histolytica* DNA editing targeting the *ehcp112* virulence gene applying the CRISPR-Cas9 methodology. The ribonucleoproteins (RNPs) formed successfully reached the nucleus and the heterochromatin, producing morphological alterations in the trophozoites. Extracts from mutant trophozoites presented higher proteolytic activity; however, the EhCP112 cysteine protease was not detected in either the secretion products or the total protein extracts. Interestingly, *ehcp112* gene knockout also altered the expression of the *ehadh* and *ehrabb* genes, also located in the V1 locus.

## 2. Materials and Methods

### 2.1. E. histolytica Cultures

Trophozoites of *E. histolytica*, strain HM1:IMSS, were axenically cultivated at 37 °C in TYI-S-33 medium (TYI-S) (Biowest. French, Rue de la Caille 49340 Nuaillé, France) and harvested during the logarithmic growth phase by subjecting the culture flasks to chilling at 4 °C [16]. Genomic DNA was isolated using the Wizard Genomic DNA Purification kit (Promega, Madison, WI, USA)) and the protein extracts were obtained in the presence of an inhibitor mixture comprising 100 mM PHMB, 100 mM PMSF, 100 mM benzamidine, 10 mg/mL aprotinin,1 µg/mL pepstatin A, 10 mg/mL leupeptin, and 1 mg/mL E-64 (Sigma, Tlanepantla, México).

### 2.2. In Silico sgRNA Design and Production

The sequence of the *ehcp112* gene (EHI_181230) was obtained from AmoebaDB (https://amoebadb.org/amoeba/app) accessed on 20 October 2022. The coding region of the gene was located between 32,770 and 34,110 base pairs (bp), and its promoter sequence was found approximately 1000 bp upstream of the ATG codon. The design of sgRNAs was performed using the Eukaryotic Pathogen CRISPR sgRNA Design Tool Software accessed on 26 October 2022 (2015 version) (grna.ctegd.uga.edu), and the single-stranded oligonucleotides were designed using the EnGen sgRNA Template Oligo Designer software accessed on 15 January 2023 (New England Biolabs) (Table 1). The T7 promoter sequence, the sgRNA, and the *S. pyogens* DNA sequence for sgRNA binding from the Cas9-scaffold was added to the templates (3′CAAAATCTCGATCTTTATCGTTCAATTTTATTCCGATCAGGCAARAGTTGAACTTTTTCACCGTGGCTCAGCCACGAAAAAA-5′). The generation of DNA templates by PCR amplification was undertaken to facilitate *in vitro* transcription, which in turn produced the mature sgRNAs [17].

### 2.3. In Vitro Transcription to Produce sgRNAs

The designed gRNAs were subjected to PCR amplification using each of the single-stranded oligos (forward primer) and the mother oligo which is the Cas9-DNA scaffold-specific sequence region for sgRNA binding (reverse primer). The forward and reverse oligonucleotides shared a 23 bp overlapping sequence. PCR was performed using 1 unit of Taq polymerase, 10 µM of sgRNA-specific forward primer, 10 µM of the reverse primer, and 1X reaction buffer. The following conditions were met in order to perform the DNA amplification: (a) 95 °C for 3 min, (b) 95 °C for 30 s, (c) 56 °C for 30 s, (d) 72 °C for 1 min, (e) steps 2 to 4 repeated 40 cycles, (f) 72 °C for 5 min. *In vitro* transcription was performed at 37 °C for 5 h using the TranscriptAid T7 kit (Thermoscientific, Waltham, MA, USA). Subsequently, the sgRNAs were purified by ethanol precipitation. Following this process, the sgRNAs were quantified using a Nanodrop 2000 instrument (Waltham, MA, USA). Thereafter, the sgRNAs were separated on 2% agarose gels in order to assess their integrity and purity grade.

### 2.4. PCR Assays

The process of PCR amplification was executed utilizing 200 ng of DNA as the template and specific primers that were designed to target the coding and promoter regions (cp-1 to cp-4) of the *ehcp112* gene (Table 2). Reactions were carried out in a volume of 50 μL using a Select Cycler (Ipswich, MA, USA) (one Taq Polymerase, New England Biolabs). The amplification cycles were comprised of the following: (a) 4 min of denaturation at 94 °C; (b) 45 cycles of 30 s of denaturation at 94 °C, 1 min of annealing at 50, 53, 56, and 60 °C, depending on the Tm of each oligonucleotide pair, and 1 min of elongation at 68 °C; (c) 5 min of elongation at 68 °C. In order to establish a negative control, the same reaction mixture was utilized, with the exclusion of a DNA template. Finally, the PCR products were analyzed by electrophoresis on 2% agarose gels and subsequently sequenced at Dr. Lorena Orozco’s laboratory at the Instituto Nacional de Medicina Genómica, Secretaría de Salud, México.

### 2.5. Production and Purification of Cas9 Endonuclease

Competent *Escherichia coli* SoluBL21 bacteria were transformed with the pET-NLS-Cas9-6xHis plasmid (Addgene, Watertown, MA, USA) and plated on LB-agar containing ampicillin at a concentration of 100 µg/mL. A single colony was selected for the inoculation of 1 L of LB medium, which had been supplemented with 100 µg/mL ampicillin until the culture reached 0.6 OD. Cas9 expression was induced by the addition of 1 mM IPTG and MgSO_4_ to the medium, and the culture was grown with constant shaking (Eppendorf, Av. Lincoln #3410 Pte, Monterrey, México) for 18 h at 18 °C. The culture was sonicated (Thermo Fisher, Av. Industria Eléctrica 7, Naucalpan de Juárez, México) for 30 s cycles, with a 5 s rest period between each cycle. Following ultracentrifugation (Beckman coulter, Gral Anaya, Benito Juarez, CDMX, México) at 40,000 rpm for 35 min at 4 °C, the resultant supernatant was filtered through a 0.22 µm pore membrane and injected into the FPLC equipment (AKTA pure system, Cytiva) (Marlborough, MA, USA). Cas9 nuclease was purified by nickel affinity chromatography column (His Trap, Cytiva), and a second purification was performed by gel exclusion chromatography. The collected fractions were run on 8% SDS-PAGE (Bio Rad, S.a. Benito Juarez, CDMX, México) to assess protein purity and integrity.

### 2.6. RNP Formation and In Vitro Assays

For the formation of RNPs, 1 µM of gRNA, 1 µM catalytically active SpCas9, 3 µL Cas9 activity buffer (10X) (New England Biolabs, Ipswich, MA, USA), and nuclease-free water (Sigma Aldrich, Toluca, México) were mixed to a final volume of 30 µL and incubated at room temperature (RT) for 15 min. Subsequently, 0.33 µM of the PCR products (cp-1 to cp-4 DNA) was added to the reaction mixture and incubated at 37 °C for 1 h. To stop the reaction, 2 µL of proteinase K (10 mg/mL) was added and the mixture was incubated for 20 min at RT. The final reaction product was subjected to electrophoresis using a 2% agarose gel (Bio Rad, S.a. Benito Juarez, CDMX, México). The obtained DNA bands were analyzed with ImageJ (V1.54K) software (NIH) to evaluate the cleavage efficiency and specificity of the gRNAs.

### 2.7. Soaking of E. histolytica Trophozoites with RNPs

Trophozoites (2 × 10^5^) were cultured in 6 mL glass tubes with serum-free TYI-33 medium at 37 °C for 1 h. The RNPs were obtained by mixing 2.5 µL of 10X activity buffer for Cas9, 10 µg of recombinant Cas9, 15 µg of sgRNA, and nuclease-free water to a final volume of 25 µL, followed by incubation at RT for 30 min. Then, 5 mL of complete TYI-S-33 medium and the previously formed RNPs were added to the trophozoites and incubated for 12, 24, 48, and 72 h at 37 °C.

### 2.8. Total RNA Extraction and RT-qPCR Assays

The isolation of total RNA from trophozoites was conducted by using the Wizard Genomic DNA Purification kit (Promega), and TRIzol reagent (Ambion by life technologies, Waltham, MA, USA), in accordance with the manufacturer’s protocols. After verifying the RNA integrity, the process of synthesizing complementary DNA (cDNA) was accomplished by the utilization of oligo dT primers and Superscript II reverse transcriptase (Invitrogen, Waltham, MA, USA). RT-qPCR (reverse transcription quantitative polymerase chain reaction) was carried out using 50 ng of cDNA; then, real-time quantitative PCR was performed using Maxima SYBR Green qPCR Master Mix (2X) (Thermo Scientific, CA, USA) in a Step One thermal cycler (Applied Biosystems, Thermo Fisher Scientific, CA, USA) under the following conditions: 50 °C, 20 min, 95 °C, 3 min, 40 cycles of 95 °C, 5 s, 66 °C, 30 s. Relative expression was calculated using the *eh40s* housekeeping gene (Table 2), and differences were determined by the the ΔΔCt method (Table 2).

### 2.9. Growth Rate Assays

In total, 1.25 × 10^5^ EhCP112-KO trophozoites and wild-type trophozoites were independently cultured in complete TYI-S-33 medium at 37 °C for 5 days. The number of trophozoites was recorded every 24 h, and their viability was assessed by trypan blue exclusion using a 0.2% dye solution, followed by examination under a light microscope (Microscopio Binocular BioBlue EBB-4260, New York, NY, USA) [18].

### 2.10. Laser Confocal Immunofluorescence

Trophozoites (cultured on coverslips) were fixed with 4% paraformaldehyde (Merck Hohenbrunn, Alemania), at 37 °C for 1 h, permeabilized using 0.2% Triton X-100 (Sigma Aldrich, Toluca, México), and blocked with 10% fetal bovine serum in PBS (Biowest. French, Rue de la Caille 49340 Nuaillé, France) and harvested during the logarithmic growth phase. Preparations were incubated at 4 °C overnight (ON) with rabbit α-EhCP112 (1:50) and mouse α-histidine (Santa Cruz, Dallas, TX, USA) (1:50) antibodies, followed by thorough washing with PBS. Thereafter, the preparations were incubated for 30 min at 37 °C with FITC-labeled α-rabbit IgGs and TRITC-labeled α-mouse IgGs (1:100). The nuclei were stained with 4′,6-diamidino-2-phenylindole (DAPI) (Sigma Aldrich, Toluca, México) and the preservation of fluorescence was achieved using Vectashield antifade reagent (Vector; Torrance, CA, USA). Preparations were examined through a Carl Zeiss LMS 700 Laser Scanning Microscope (Carl Zeiss Microimaging GmbH, Jena, Alemania) confocal microscope, in 0.5 µm laser sections, and images were processed with ZEN 2009 Light Edition Software Edition 5.5 (Zeiss; Jena, Germany).

### 2.11. Transmission Electron Microscopy (TEM)

The samples were prepared for TEM as described [18]. In summary, trophozoites were fixed with 4% paraformaldehyde and 0.5% glutaraldehyde in PBS for 1 h at RT. Samples were embedded in LR White resin (London Resin Co., Enfield, UK) (Electron Microscopy Sciences, Hatfield, PA, USA) and polymerized under UV light (Crisol, Guadalajara, México) at 4 °C for 48 h to obtain thin sections (60 nm) that were mounted on Formvar-covered nickel grids followed by ON incubation with rabbit α-EhCP112 (1:20) and mouse α-histidine antibodies (Santa Cruz, Dallas, TX, USA) (1:20). Thin sections were incubated ON with gold-labeled (15 nm) conjugated α-mouse antibodies (for α-histidine) (1:50) or gold-labeled (30 nm) α-rabbit antibodies (for α-EhCP112) (1:50), then the samples were contrasted with uranyl acetate and lead citrate and observed through a Joel JEM-1400 (JEOL, Tokyo, Japan) transmission electron microscope.

### 2.12. Scanning Electron Microscopy

The trophozoites were fixed with 2.5% (v/v) glutaraldehyde (Sigma Aldrich, Toluca, México) in 0.1 M sodium cacodylate buffer (Thermo Fisher Scientific, MA, USA) pH 7.2 for 1 h at RT, dehydrated with increasing concentrations of ethanol, and CO_2_ critically point dried in a Samdri apparatus (Tousimis Corp., Rockville, MD, USA). Then, the cells were gold coated using an ion sputtering device (TED Pella, Redding, CA, USA) and observed through a JEOL-JSM 6510 LV scanning electron microscope (JEOL Ltd., Tokyo, Japan).

### 2.13. Western Blot Assays

The total extracts from trophozoites (prepared with the protease inhibitors cocktail), were separated by SDS-PAGE gels, transferred onto nitrocellulose membranes, and probed with rabbit α-EhCP112 (1:5000), rabbit α-EhRabB (1:2000) [19], rat α-EhVps23 (1:500) [20], and α-human actin (1:2000) (Santa Cruz Biotechnology) (Dallas, TX, USA) antibodies. Membranes were incubated with species-specific horseradish peroxidase (HRP) (Merck KGaA, Darmstadt, Alemania)-labeled secondary antibodies (1:10,000) and developed with the ECL Prime detection reagent (GE-Healthcare) (TCR Scientific, CDMX, México), following the manufacturer’s instructions on a MicroChemi System (DNR Bio-Imaging) (Eppendorf, Hamburg, Germany).

### 2.14. Protease Activity

The EhCP112-KO trophozoites and the wild-type trophozoites (3 × 10^6^) were harvested and washed with sterile PBS by centrifugation at 1700 rpm for 7 min at 4 °C. The supernatant was then discarded, and the pellets were resuspended in 200 µL of PBS and incubated for 2 h at 37 °C to obtain the secretion products [21]. The proteolytic activity of the secretion products was analyzed in 0.2% polyacrylamide gelatin gels (Merck KGaA, Darmstadt, Alemania) as described [22]. Following electrophoresis, gels were incubated in 2.5% Triton X-100 for 1 h at RT. Enzymatic activity was activated by incubating the gels in 0.1 M Tris-HCl ( Sigma Aldrich, Toluca, México), pH 7 (activation buffer) for 16 h at 37 °C. The gels were stained with 0.25% Coomassie blue R-250 ) (Merck KGaA, Darmstadt, Alemania) for 30 min and destained with a mixture of 10% acetic acid and 50% methanol. The presence of clear bands indicative of proteolytic activity was observed. Replicates of the same experiment were prepared for the purpose of performing Western blot assays for EhCP112 recognition.

### 2.15. Statistical Analyses

All experimental values are presented as the mean and standard error from at least three independent assays, carried out by duplicate. Statistical analyses were conducted using the GraphPad Prism v5.01 software by a paired Student’s t test. * *p* < 0.05; ** *p* < 0.01, and *** *p* < 0.001.

## 3. Results

### 3.1. sgRNA Design and Production

Figure 1A shows the schematic of the location of the three genes that conform the V1 locus. In order to achieve a knockout of the *ehcp112* gene, ten cp-gRNAs candidates were designed, each of which contained the required PAM sites for DNA editing. The four highest-scoring cp-gRNAs were selected based on the algorithm provided by the Eukaryotic Pathogen CRISPR sgRNA Design Tool accessed on 26 October, 2022 (2015 version) (Table 1). Two of the selected cp-gRNAs were located in the promoter region and the other two in the gene’s ORF (Figure 1B). After obtaining the DNA templates carrying the sgRNA sequences (Table 1), the cp-gRNAs were transcribed using as primers the oligonucleotides corresponding to the designed cp-gRNAs and the Cas9-DNA specific sequence scaffold region. The resulting DNA templates were then used to obtain the mature cp-gRNAs. As a control, we used the sgRNA corresponding to the green fluorescent gene (Figure 1C–E).

### 3.2. The ehcp112 Gene Was In Vitro Edited Using the CRISPR-Cas9 System

The results obtained by Mendes Kangussu-Marcolino et al. (2021) [7] evidenced that the CRISPR-Cas9 strategy is functional in *E. histolytica* trophozoites for editing an exogenous luciferase gene. However, the authors did not manipulate *E. histolytica* DNA. In this study, the system efficiency was probed using *in vitro* formed RNP complexes composed of the *S. pyogenes* Cas9 enzyme and the synthesized mature cp-gRNAs sequences (Figure 2A,B). *In vitro* editing was carried out in *E. histolytica* DNA (Figure 2C). The four target regions containing the cp-1, cp-2, cp-3, and cp-4 sequences were PCR-amplified (Table 2, Figure 2D) to evaluate the RNPs’ cleavage efficiency on the amplified regions. The results obtained demonstrated that not all cp-gRNAs were equally efficient to produce *in vitro* DNA editing (Figure 2B,E), while the cp-3 and cp-4 gRNAs efficiently cleaved the corresponding DNA regions,, no effect was observed when cp-1-gRNA and cp-2-gRNA were used (Figure 2E,F). Moreover, cp-3-gRNA showed higher DNA cleavage efficiency (~100%), whereas cp-4 gRNA presented only about 65% efficiency (Figure 2G). Remarkably, these experiments disclosed that the CRISPR-Cas9 system is effective for *E. histolytica* in *in vitro* DNA editing (Figure 2).

### 3.3. The RNPs Formed by the Cas9 Enzyme and cp-gRNAs Are Efficiently Introduced into the Cytoplasm and Nucleus of E. histolytica Trophozoites

Subsequently, the efficiency of the CRIPSR-Cas9 system edition of *E. histolytica* genes was validated. For this to occur, the following conditions must be taken into consideration: (i) the RNPs formed by Cas9 and the corresponding cp-gRNAs must penetrate the trophozoites’ plasma membrane, reach the nucleus and access the chromatin; (ii) the enzyme and the cp-gRNAs must not damage the trophozoites; (iii) the target gene must be modified (Figure 3).

To form the RNPs, the selected cp-3- and cp-4-gRNAs were independently incubated with the purified recombinant Cas9 enzyme for 30 min at RT, as described in the Section 2. Then, the trophozoites (previously fasted for one hour) and the RNP complexes were soaked. At different incubation times, the intracellular pathway followed by the RNPs was monitored using α-histidine antibodies that recognize the histidine tail added to the recombinant Cas9 enzyme. Simultaneously, the gene expression of *ehcp112* in RNP-treated cells (EhCP112-KO trophozoites) was assessed using the α-EhCP112 antibodies [23]. In the HM1:IMSS trophozoites, confocal laser microscopy confirmed the presence of abundant EhCP112 protein in the plasma membrane and in cytoplasmic small vesicles (Figure 4). Additionally, some scarce marks were detected in the nucleus. As expected, no fluorescence signals were detected when we used the α-histidine antibodies. In contrast, the EhCP112-KO trophozoites exhibited reduced levels of EhCP112 protein, showing scarce fluorescent marks (Figure 4), but the Cas9 enzyme appeared in the plasma membrane, cytoplasm, and around and inside the nucleus of mutant trophozoites. DAPI was used as a DNA marker, colocalized in the nuclei with the α-histidine antibodies (Figure 4). The remnant fluorescence displayed in the EhCP112-KO trophozoites by α-EhCP112 antibodies may be due to the protein synthesized before the CRISPR-Cas9 treatment. These results strongly suggest that the RNPs penetrated the EhCP112-KO trophozoites’ plasma membrane and reached the nucleus.

### 3.4. Transmission Electron Microscopy (TEM) Evidenced That the RNPs Reached the Chromatin in E. histolytica Trophozoites

To further confirm that the RNPs had actually reached the nucleus and contacted the chromatin of the EhCP112-KO trophozoites, TEM experiments were performed. For this purpose, 60 nm thin slides of EhCP112-KO trophozoites were incubated with mouse α-histidine antibodies, and then, with 15 nm gold-labeled α-mouse antibodies. Preparations were also incubated with rabbit α-EhCP112 antibodies and 30 nm gold-labeled secondary α-rabbit antibodies. The TEM images confirmed that the Cas9 recombinant enzyme had reached the cytoplasm and the nucleus (Figure 5A,B). Interestingly, the 15 nm gold-labeled antibodies were also located in the chromatin (Figure 5B), thereby providing compelling evidence of the interaction between RNPs and DNA. The immunological profile of the α-EhCP112 antibodies exhibited paucity of reactivity in mutant trophozoites but appeared in the nucleus (Figure 5B). In contrast, the HM1:IMSS trophozoites showed abundant 30 nm gold particles, which have been associated with the plasma membrane, cytoplasmic vesicles and extracellular space (Figure 5C), confirming that EhCP112 is secreted by trophozoites [21]. As expected, no α-histidine marks appeared in the wild-type HM1:IMSS cells. As negative controls, we used gold-labeled secondary antibodies—omitting the primary antibodies (Figure 5C). At this time, the function of the EhCP112 protein in the nucleus remains to be elucidated. Several reports in different systems have confirmed the presence of cysteine proteases in the nucleus [24]. The authors propose that these enzymes probably do not participate directly in transcription or translation, but rather have a regulatory role in gene expression [24]. Altogether, our results strongly suggest that *E. histolytica* DNA can be edited by an *in vivo* CRISPR-Cas9 system, since the enzyme reached the nuclei and the chromatin, while *in vitro* assays confirmed that the RNPs knocked out the target gene (Figure 1 and Figure 2), demonstrating for the first time the efficiency of the system in *E. histolytica* trophozoites.

### 3.5. EhCP112-KO Trophozoites Grew at a Similar Rate than the HM1:IMSS Strain, but They Presented Changes in Their Morphology

To explore the possible damage produced in the trophozoites by the treatment, the viability of the trophozoites was measured after incubation for different times with the RNPs. In all cases, the exclusion of trypan blue was observed in 95% of the EhCP112-KO trophozoites, thereby indicating their viability. No substantial disparities were observed in the growth rate of the wild type and the mutant populations (Figure 6A), confirming that no significant damage had been produced by the RNPs. To corroborate the morphology of the cells after incubation of the trophozoites with the RNPs, we analyzed them through a scanning electron microscope (SEM). SEM images revealed that the EhCP112-KO trophozoites exhibited a slightly rounded morphology, reduced in size and with fewer pseudopodia compared to the wild-type cells. However, their morphology strongly suggests that they were alive at the time of glutaraldehyde fixation (Figure 6B). Furthermore, the majority of the mutated cells exhibited depressions in their plasma membrane. The functional implications of these morphological alterations in EhCP112-KO trophozoites remain to be elucidated. Therefore, further research is required to provide a detailed answer in this regard.

### 3.6. The ehcp112 Gene Sequence Is Altered and the Gene Is Not Expressed in the EhCP112-KO Trophozoites

PCR assays evidenced the presence of the *ehcp112* gene in both mutant and wild- type trophozoites. As expected, no appreciable differences in the migration of PCR-amplified fragments from either population were discernible in the 2% agarose gels (Figure 7A), given the limited variation between the genes, which is confined to a few nucleotides. However, the results of the RT-qPCR assays, using the designed oligonucleotides (Table 2), evidenced that the *ehcp112* gene was expressed in the HM1:IMSS trophozoites, but not in the RNP-treated trophozoites (EhCP112-KO) (Figure 7B). Sequencing of the PCR-amplified fragments (Figure 7A) showed that four nucleotides (TCTT), adjacent to the selected PAM site, were absent in the mutant gene (Figure 7C), precisely at the region where the cp-3-gRNA hybridized with the DNA as shown in the scheme depicted in Figure 7D. These results strongly evidenced the functioning of the modified CRISPR-Cas9 strategy implemented *in vivo* for DNA editing.

### 3.7. The EhCP112 Protein Is Expressed in HM1:IMSS but Not in EhCP112-KO Trophozoites

To fully understand the function of a given protein, it is required to obtain mutant cells in which the product of the edited gene is absent or at least mutated to produce a non-functional protein. (Figure 8A). We searched for the EhCP112 protein in the EhCP112-KO trophozoites by Western blot analysis, using α-EhCP112 antibodies. The band corresponding to the EhCP112 protein appeared in the HM1:IMSS trophozoites, as well as in the controls independently incubated only with Cas9 or sgRNA (Figure 8A). However, after 24 h incubation of the cells with the RNPs, the EhCP112 protein was no longer detected by Western blot assays, either when trophozoites were incubated with cp-3-gRNA (located in the promoter and exhibiting ~100% DNA cleavage efficiency) or cp-4-gRNA (located in the ORF and exhibiting 65% DNA cleavage efficiency) (Figure 8A). In all experiments the wild-type phenotype was recovered 48 h after incubation with the RNPs. We are currently working on obtaining stable mutants for longer times using cloned populations. In some experiments, multiple bands were detected by the antibodies on the nitrocellulose membranes. These bands correspond to the protein zymogens (52 kDa, the prepropeptide; 50 kDa, the propeptide; and 43 kDa, the mature enzyme). The gene editing produced by the two cp-gRNAs modified the protein expression by knocking out the *ehcp112* gene.

### 3.8. The ehcp112 Gene Knockout Augments the Enzymatic Activity of EhCP112-KO Trophozoite Extracts but Its Activity Diminished in the Secretion Products

*E. histolytica* is known to possess approximately 50 cysteine proteases. Some of them have been characterized [25]. The present study explored the activity of total extracts and secretion products from both mutant and wild-type trophozoites. First, lysates were separated on 12.5% SDS-PAGE-gelatin substrate to detect the gelatin digestion produced by proteases. Intriguingly, total extracts from EhCP112-KO trophozoites exhibited a greater capacity to digest gelatin gels, than those from HM1:IMSS lysates, suggesting that the *ehcp112* knockout activated other proteases (Figure 8B), as has been reported for other systems [26]. To investigate the absence of EhCP112 in the EhCP112-KO trophozoites, we performed Western blot assays in parallel. The α-EhCP112 antibodies revealed a single 50 kDa band only in the HM1:IMSS extracts. As a loading control, we used α-actin antibodies (Figure 8B–D). Given that the EhCP112 cysteine protease is secreted [21], we next explored its presence in the secreted products of both populations. The gelatin gels evidenced that, in contrast to the total extract assays, the proteinase activity in the secretion products was higher in the HM1:IMSS trophozoites than in the EhCP112-KO trophozoites (Figure 8E). However, we also observed white bands in the lane containing the secreted products obtained from the mutant cells, suggesting the activity of other proteases in the EhCP112-KO trophozoites as Western blot assays indicated the absence of EhCP112 (Figure 8F). As a positive secretion control, we used the α-EhVps23 antibodies, and as a cell integrity control, we used α-actin (Figure 8G,H).

### 3.9. The ehcp112 Gene Knockout Affects Other Genes of the V1 Locus

Finally, we examined the effect of the *ehcp112* gene knockout on other members of the V1 locus. Interestingly, the RT-qPCR assays evidenced a strong decrease in the expression of the *ehadh* and *ehrabb* genes in the knocked-out trophozoites (Figure 9A). Western blot assays confirmed that the EhADH and EhRabB protein expression was also affected (Figure 9B). While no bands were detected with the α-EhADH antibodies, the α-EhRabB antibodies detected a significant diminishing of the EhRabB protein at 12 and 24 h, as evidenced in the densitometry graphs (Figure 9C).

## 4. Discussion

In the present paper, we report the application of the CRISPR-Cas9 strategy to edit *E. histolytica* genes for the first time. Evidence was also provided that the knockout of the *ehcp112* gene has an impact on the *ehadh* and *ehrabb* genes of the V1 virulence locus. The relevance of this work lies in the following: (i) the implementation for the first time of the CRISPR-Cas9 strategy in *E. histolytica* trophozoites; (ii) the simplification of the CRISPR-Cas9 methodology for gene editing that minimizes the damage to the trophozoites during procedures; (iii) the strong experimental suggestion of the coordinated expression of the V1 locus genes. The modifications that were carried out on the conventional CRISPR-Cas9 strategy for gene knockout will facilitate the study of molecules involved in the aggressive mechanisms seen with a high survival of the parasites. At the same time, it opens access for the design of new strategies to combat amoebiasis.

Most of the papers reporting the utilization of the CRISPR-Cas9 strategy for gene editing employ an expression vector carrying the Cas9 gene and the sgRNA cassette with the U6 RNA polymerase III promoter [27]. This plasmid is then transfected into the target cell line [28]. In this work, we used a simplified CRISPR-Cas9 system that consists of two components: (i) the purified Cas9 protein; (ii) the *in vitro* transcribed sgRNAs from a DNA template obtained by PCR amplification.

This methodology enabled the verification of the specificity and DNA cleavage efficacy of the RNPs *in vitro*, prior to the conduction of the *in vivo* assays. In this study, not all RNP complexes were equally efficient in DNA cleavage. The cp-3-gRNA was selected for the majority of the experiments presented here due to its high score, according to the algorithm of the Eukaryotic Pathogen CRISPR gRNA Design Tool. It renders ~100% cleavage efficiency and proved to be specific in the *in vitro* experiments. The differences shown by the sgRNAs could be attributable to the following: (i) steric hindrance in the DNA that prevents its interaction with RNPs; (ii) a weak pairing of gRNA with the target. This latter may be due to the sequence of the regions being obtained from the gene annotated in the AmoebaDB database and because the HM1:IMSS strain may have suffered DNA changes over more than 30 years of culture and several passages through animal livers to regain virulence; (iii) conformational rearrangements that avoid Cas9 nuclease activation prior to DNA editing [29]. For these experiments, the Cas9 enzyme, the sgRNA, and the RNPs were synthesized *in vitro*; thus, this method can cause a lower toxicity in the EhCP112-KO trophozoites after gene editing, when compared to the use of the enzyme produced by the cell machinery as is the case in other systems [30]. Furthermore, the *in vitro* synthesized gRNAs could also be less toxic and have greater specificity, because they can be tested *in vitro*, and then, the RNPs can be introduced into the cell to directly reach the selected target, increasing the editing efficiency [31].

The RNPs were first probed *in vitro*. This approach reduces the probability of toxic effects on the cell and prevents off-target editing [32]. In addition, to introduce RNPs into the trophozoites, we used a non-aggressive technique that consisted in trophozoites soaking [33] with the RNPs without using electroporation—which kills many cells in transfection experiments. In contrast, the simple soaking does not seem to produce significant damage to the cells [34]. The confocal and TEM images showed that the Cas9 enzyme, as a part of the RNPs, reaches the nuclei and the heterochromatin. Growth curves and trypan blue exclusion assays showed that the cells remained viable through the experiments. This was corroborated by SEM observations. Images revealed trophozoites displaying the typical morphology of living cells, except for the membrane depressions, which did not affect cell viability. The causes of these phenotypic changes and their physiological effects remain unknown.

The PCR-amplified region sequences containing part of the *ehcp112* gene promoter where the cp-3 gRNA hybridizes showed the deletion of four base pairs in the gene. In addition, other nucleotide impairments were observed in the mutant gene. Some of these mismatches may have occurred during DNA repair or as an effect of the assays, since they do not appear in the HM1:IMSS trophozoites. The RNP-induced mutation resulted in a strong decrease in *ehcp112* gene expression, as demonstrated in RT-qPCR experiments.

Once the deletion was verified by DNA sequencing, we proceeded to explore the effect of gene editing on protein expression. In our conditions, 24 h was determined to be the best time to knock out the target gene. The effect obtained in these experiments lasted only 24 h, because after 48 h the trophozoites recovered EhCP112 expression, probably due to the fact that these experiments were carried out with the old and heterogeneous HM1:IMSS strain and due to DNA repair by homologous recombination as well as other events occurring in the heterogeneous culture [35]. This is the first time that a given gene has been knocked out in *E. histolytica* for 24 h. This will allow the study of the precise role of the EhCP112 protein over the course of this period. The strain employed in this study was heterogeneous, formed by trophozoites with some individual differences. Probably, this CRISPR-Cas9 strategy produced an effect in the majority of the trophozoites, but not all. Therefore, further research will try to elucidate this using clone A (daughter cell line of the HM1:IMSS strain) [3]. However, this clone was obtained several decades ago and it is necessary to clone it again. Thus, we require more time in order to obtain a homogeneous population and characterize these cells before performing the DNA editing experiments.

Intriguingly, substrate gel experiments exhibited a higher protease activity in the EhCP112-KO than in the HM1:IMSS trophozoites. *E. histolytica* has a high number of proteases and other enzymes that must be activated, producing strong digestion in the gelatin gels, as has been reported for other systems [26]. The presence of EhCP112 in the nucleus of trophozoites suggests that this enzyme has an unknown role in nuclear functions. Thus, we hypothesize that EhCP112 could have a regulatory role for the expression of other proteases. Therefore, further experimental evidence is required to substantiate this hypothesis. Nevertheless, the enzyme was not detected in Western blot assays either in the total extracts or in the secretory products of the EhCP112-KO trophozoites, confirming that the protease activity observed in the gelatin gels was not due to EhCP112.

Previous studies have demonstrated that *ehcp112* gene knockdown, as well as the utilization of siRNAs, affects *ehadh* gene expression [36]. Here, deleting only 4 bases in *ehcp112*, the gene resulted in a knockout. The distance between the cp-3-gRNA target and the *ehadh* initiation codon is 1840 base pairs. The *ehadh* gene has a distal enhancer element that is characterized at −996 bp upstream; it has a putative silencer at −766 bp and a minimal promoter at −150 bp from the ATG [37]. The cp3-gRNA hybridizes at 502 bp of the *ehcp112* ORF that overlaps with the *ehadh* promoter, explaining the *ehadh* knockout. In addition, its promoter could have distant motifs that could be affected by the RNPs. This could be a working hypothesis to study the unknown mechanism that influences *ehadh* gene expression. The reduction in *ehrabb* gene expression can be explained by the fact that its ORF overlaps with the cp-3-gRNA localization. It is acknowledged that this work has given rise to several unresolved issues. It is anticipated that future research will provide some answers.

## Figures and Tables

**Figure 1 microorganisms-13-02219-f001:**
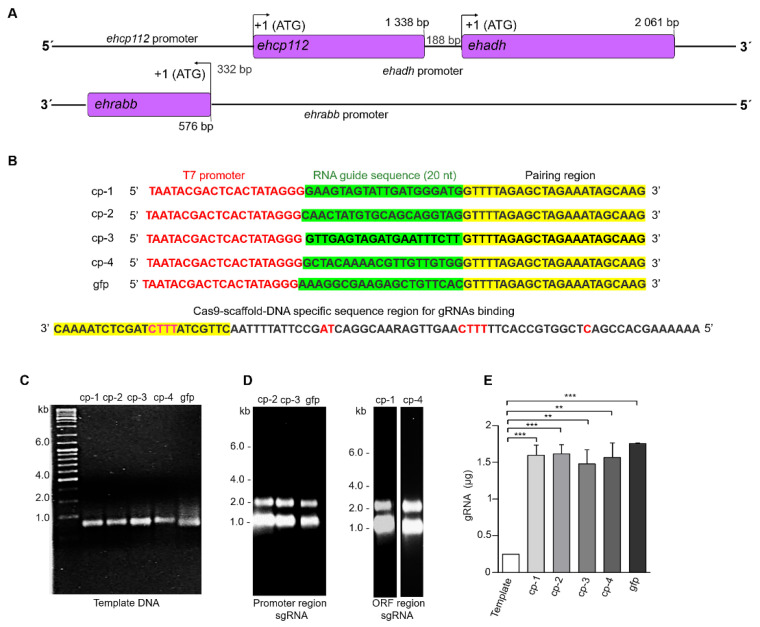
Design and production of the sgRNAs. (**A**): Scheme of the V1 virulence locus. Arrows: ATG initiation codon. Numbers: Location of genes and promoters. (**B**): DNA templates containing the T7 promoter sequences, the RNA guide sequence, and the region for the Cas9 pairing sequences. On the left: Names of the synthesized sgRNAs. At the bottom: CRISPR DNA conserved sequence for sgRNAs binding. (**C**): Agarose gels (2%) with the amplified DNA templates of the four selected sgRNAs and the sgRNA for the fluorescent green protein (gfp) used as a control. (**D**): Transcripts corresponding to the sgRNAs located in the promoter region (cp-2 and cp-3) and the open reading frame (ORF) (cp-1 and cp-4) region. (**E**): Production efficiency (RNA µg) for each of the sgRNAs shown in (**D**) ** *p* < 0.01 and *** *p* < 0.001.

**Figure 2 microorganisms-13-02219-f002:**
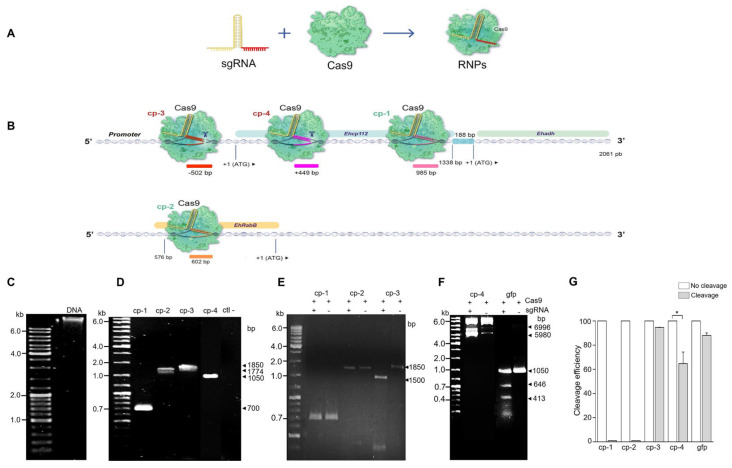
*In vitro ehcp112* gene editing using the CRISPR-Cas9 strategy in *E. histolytica.* (**A**): Scheme of the sgRNA and the Cas9 to form the RNPs complexes. (**B**): Scheme illustrating the predicted position of the RNPs in the *ehcp112* gene. As depicted in the DNA scheme, the *ehcp112*, *ehadh,* and *ehrabb* genes are represented. Numbers: Regions where RNPs binds in the V1 virulence locus. Scissors: Cleavage sites. (**C**–**F**): Products obtained following RNP treatment analyzed in 2% agarose gels. (**C**): Genomic DNA of *E. histolytica*. (**D**): PCR-amplified products of the regions containing cp-1, cp-2, cp-3, cp-4, and negative control (ctl-). (**E**,**F**): cp-1, cp-2, cp-3, cp-4, and gfp transcripts after interaction with the RNPs as described in the Section 2. (**G**): Densitometry of the RNP cleavage efficiency quantification * *p* < 0.05.

**Figure 3 microorganisms-13-02219-f003:**
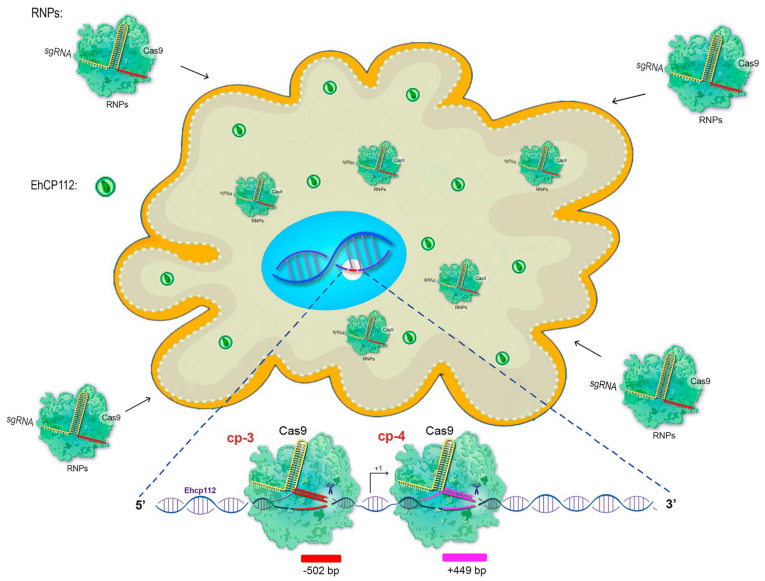
Cartoon of the *in vivo* CRISPR-Cas9 system predicting the knockout of the *ehcp112* gene. The scheme shows hypothetical results after soaking the trophozoites with the RNPs that efficiently cleavage the DNA. Bottom: Scheme of the predicted DNA cleavage sites assuming that the RNPs pass through the plasma and nuclear membranes in order to reach the target gene.

**Figure 4 microorganisms-13-02219-f004:**
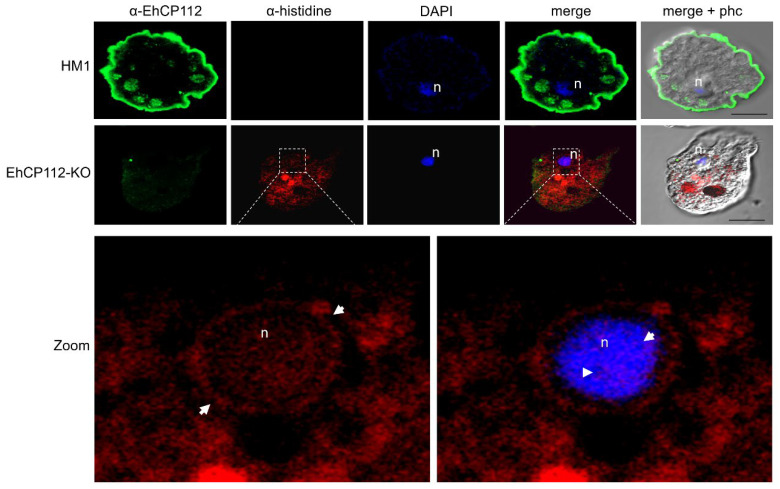
Confocal laser microscopy of the RNPs inside the cell and the nucleus. Images of the HM1:IMSS trophozoites and the EhCP112-KO trophozoites labeled with α-EhCP112 antibodies (green) and α-histidine antibodies (red), to track the RNPs inside the cell. n: Nuclei stained by DAPI (blue). Zoom: Magnification of the EhCP112-KO nucleus region marked with square dotted lines in the upper panel. Arrows: RNPs in the nucleus and the nucleus periphery. Right panel: DAPI and α-histidine merging.

**Figure 5 microorganisms-13-02219-f005:**
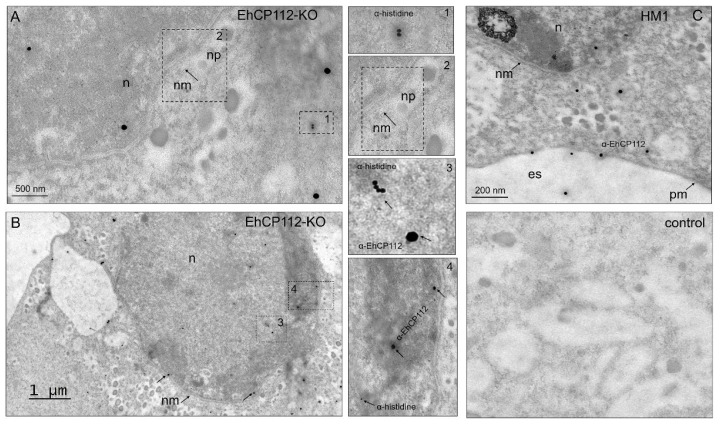
TEM of the RNPs inside the cell, in the nucleus, and in the heterochromatin. The EhCP112-KO trophozoites and HM1:IMSS trophozoites were embedded in white resin. Then, 60 nm slides were incubated with mouse α-histidine followed by α-mouse gold-labeled (15 nm) antibodies and rabbit α-EhCP112 followed by α-rabbit gold-labeled (30 nm) antibodies, contrasted with uranyl acetate and lead citrate, as described in the Section 2. (**A**,**B**): EhCP112 and the RNPs in the nucleus (n) (**A**) and in the heterochromatin (**B**). nm: Nuclear membrane, np: Nuclear pore. Dotted squares: magnification in 1, 2, 3, 4 squares. (**C**): HM1:IMSS trophozoites showing the EhCP112 protein in cytoplasm, nucleus (n), plasma membrane (pm), in the extracellular space (es) and the arrows point to the EhCP112 and histidine proteins. Control: Trophozoites incubated only with the secondary antibodies.

**Figure 6 microorganisms-13-02219-f006:**
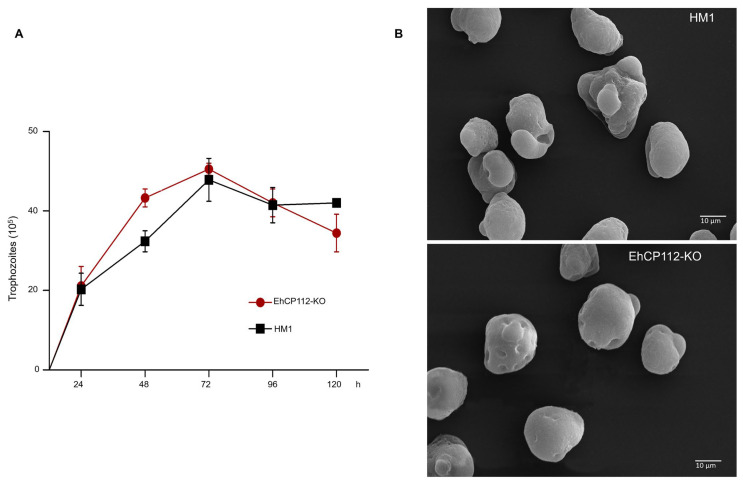
Growth curve and scanning electron microscopy of the EhCP112-KO trophozoites. (**A**): Trophozoites were incubated in TYI-S-33 medium for different times, and the number of cells was counted each 24 h. (**B**): Control and mutant trophozoites were glutaraldehyde-fixed and treated for SEM as described in the Section 2. The cells showed membrane depressions in the EhCP112-KO trophozoites.

**Figure 7 microorganisms-13-02219-f007:**
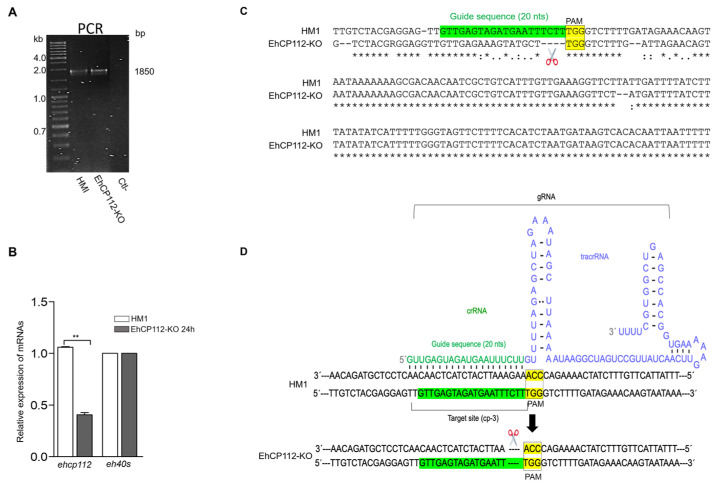
EhCP112-KO gene editing. (**A**): PCR of the 1850 bp containing the edited region and negative control (Ctl-). (**B**): RT-qPCR of the *ehcp112* gene in the EhCP112-KO trophozoites after 24 h RNPs treatment. The *eh40s* gene was used as a control ** *p* < 0.01. (**C**): DNA sequence of the *ehcp112* gene fragment containing the edited region (scissors). sgRNA sequence: Green. PAM sequence: Yellow. Points: Changed bases in the mutant. Asterisks: Conserved bases in both populations. (**D**): Scheme representing the secondary structure of the gRNA in hybridization with the DNA of the HM1:IMSS trophozoites to produce the mutation.

**Figure 8 microorganisms-13-02219-f008:**
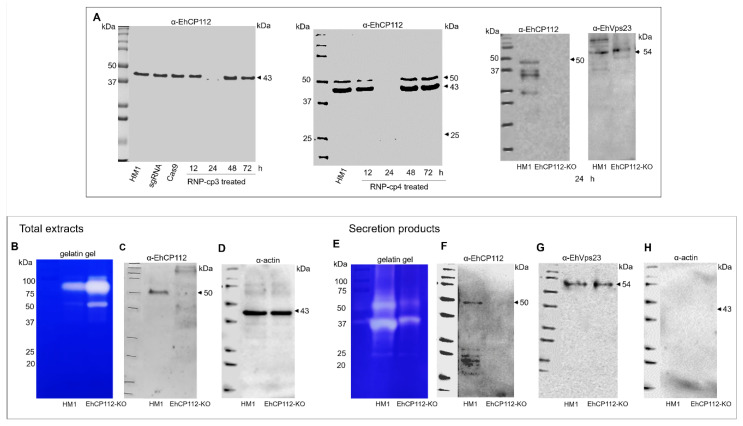
Western blot assays and protease activity of the HM1:IMSS and EhCP112-KO trophozoites. Trophozoites were lysed in the presence of an inhibitor mix as described in the Section 2. (**A**): Total proteins separated in 12.5% SDS-PAGE and transferred onto nitrocellulose membranes, incubated with rabbit α-EhCP112 antibodies followed by α-rabbit-HRP-labeled antibodies. Upper panel: Trophozoites treated with RNP-cp-3. Bottom panel: Trophozoites treated with RNP-cp4. (**B**): Gelatin gel of total extracts of the HM1:IMSS and the EhCP112-KO trophozoites. (**C**): Parallel Western blot of proteins in (**B**) using α-EhCP112 antibodies and the secondary α-rabbit-HRP-labeled antibodies. (**D**): Western blot using mouse α-actin and α-mouse HRP-labeled antibodies as loading control. (**E**): Gelatin gel of secretion products of theHM1:IMSS and the EhCP112-KO trophozoites. (**F**): Parallel Western blot of secretion products from the HM1:IMSS and the EhCP112-KO trophozoites. (**G**): Parallel Western blot using α-EhVps23 as a secretion control. (**H**): Parallel Western blot using α-actin as a cellular integrity control.

**Figure 9 microorganisms-13-02219-f009:**
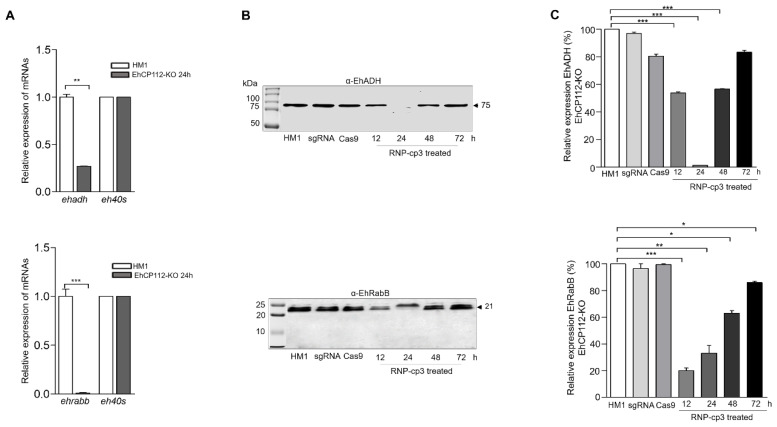
The *ehadh* and *ehrabb* genes were knocked out as an effect of the CRISPR-Cas9 trophozoite treatment. (**A**): RT-qPCR of *ehadh* (**upper**) and *ehrabb* (**bottom**). The *eh40s* gene was used as a control. (**B**): Trophozoites lysed in the presence of an inhibitor mix as described in the Section 2. and transferred onto nitrocellulose membranes. Membranes were cut and the upper part of the membrane was incubated with α-EhADH antibodies and the bottom part with α-EhRabB antibodies followed by the respective HRP-labeled secondary antibodies. (**C**): Densitometry of bands in (**B**) * *p* < 0.05, ** *p* < 0.01 and *** *p* < 0.001.

**Table 1 microorganisms-13-02219-t001:** cp-gRNA sequences containing the PAM sites.

gRNA	Sequence (PAM “NGG”)	Location	% GC	Off Target	Score
cp-1	GAAGTAGTATTGATGGGATG**GGG**	ORF-985 bp	40%	no	0.62
cp-2	CAACTATGTGCAGCAGGTAG**TGG**	Promoter 662 bp reverse	50%	no	0.61
cp-3	GTTGAGTAGATGAATTTCTT**TGG**	Promoter 502 bp	30%	no	0.59
cp-4	GCTACAAAACGTTGTTGTGG**TGG**	ORF-449 bp	45%	no	0.58
cp-5	AATGGCGGTACTTCATTCCA**TGG**	ORF-337 bp reverse	45%	no	0.57
cp-6	GCAGGATCTGACTTTCTCAT**TGG**	Promoter 244 bp	45%	no	0.56
cp-7	GAAGTACCGCCATTACCTTC**TGG**	ORF-364 bp	50%	no	0.56
cp-8	GGGTGTTGTTGCATGGCTCT**AGG**	Promoter 539 bp reverse	55%	no	0.56
cp-9	GTTCTGAAGTGCATAGGTAA**AGG**	Promoter 281 bp	40%	no	0.56
cp-10	AAATTCATGGGCTAGTGGAT**GGG**	Promoter 480 bp	40%	no	0.56

Score: As the score approaches 1, the probability of hybridization with target DNA increases.

**Table 2 microorganisms-13-02219-t002:** Primers used to amplify the DNA regions via PCR.

Primers	Sequence	Assay
cp-1 Forward	5′ CAAAACGTTGTTGTGGTGGTC 3′	PCR
cp-1 Reverse	5′ TGATTGTAGAATTGGACATAGGTTG 3′	PCR
cp-2 Forward	5′ TGCCTTTACCTATGCACTTCAGA 3′	PCR
cp-2 Reverse	5′ TGCCACTCTAAGTCGTTGGAC 3′	PCR
cp-3 Forward	5′ AACACCTTCACCAGTTTTGGC 3′	PCR
cp-3 Reverse	5′ CCCATCCACTAGCCCATGAA 3′	PCR
cp-4 Forward	5′ CAGCGATTGTTGTCGCTTTTT 3′	PCR
cp-4 Reverse	5′ CCCATCCACTAGCCCATGAA 3′	PCR
cp112-Forward	5′ GGAGGTTGTTGGGCAGTTTC 3′	RT-qPCR
cp112-Reverse	5′ CTTCACCCCATCCACTAGCC 3′	RT-qPCR
Eh40s-Forward	5′ ATTCGGAAATAGAAGAGGAGG 3′	RT-qPCR
Eh40s-Reverse	5′ ACTAATCTTCCAAGCTTGGT 3′	RT-qPCR

## Data Availability

The original contributions presented in this study are included in the article. Further inquiries can be directed to the corresponding author.

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
