# Peer review of "The CRISPR-Cas9 System in Entamoeba histolytica Trophozoites: ehcp112 Gene Knockout and Effects on Other Genes in the V1 Virulence Locus"

_microorganisms, 2025, doi:10.3390/microorganisms13092219_

Round 1
Reviewer 1 Report
Comments and Suggestions for Authors
This manuscript reports the first successful editing and knockout of the ehcp112 gene in Entamoeba histolytica (E.h) using the CRISPR-Cas9 system. By designing and synthesizing specific sgRNA and expressing the Cas9 enzyme, precise editing of the target gene was achieved, evidenced by changes in the gene sequence and the absence of the target protein. This study not only demonstrates the application potential of CRISPR-Cas9 technology in E.h but also provides a foundation for understanding the function of the ehcp112 gene and its role in the life cycle of E.h. Additionally, by analyzing phenotypic changes in edited E.h, such as cell morphology, growth rate, and protein expression levels, researchers explored the potential role of the ehcp112 gene in the invasiveness and pathogenicity of E.h, paving the way for further elucidation of its biological functions. This study provides methodological support for the development of new therapies and vaccines against E.h infection and also highlights the experimental details and potential challenges to be noted when applying CRISPR-Cas9 technology for gene editing. Overall, the manuscript is well-written and clearly articulated, making it suitable for publication in Microorganisms with minor revisions. The following points should be addressed before publication:
- in abstract, the authors should clearly point out the novelty of this work.
- The referencing style appears inconsistent. The authors should standardize the referencing format throughout the manuscript to ensure a professional presentation.
Author Response
Dear Reviewer: Thank you very much for your very kind comments. We have highlighted our findings in the abstract and we have uniformed the refences according to the journal format. Please find attached our new version that has been significantly improved with your comments.
Reviewer 1
- in abstract, the authors should clearly point out the novelty of this work.
R= Recommendation followed.
- The referencing style appears inconsistent. The authors should standardize the referencing format throughout the manuscript to ensure a professional presentation.
R= Recommendation followed.
Reviewer 2 Report
Comments and Suggestions for Authors
Dear authors,
The entry manuscript (MS) “The CRISPR-Cas9 system in Entamoeba histolytica trophozoites: ehcp112 gene knockout and effects in other genes of the V1 virulence locus” fits very well in the in the scope of the special volume “Advances in Molecular Biology of Entamoeba histolytica” of Microorganisms from MDPI. However, there are so many points to be solved before the final approval of MS. The major issue constitutes the efficacy of your ehcp112 knockout, since the absence of the protein was just validated 24 hours after CRISPR/Cas9 edition. The addition of a resistance marker, or at least a clonal selection, should be done to obtain 100% knockout parasites. CRISPR/Cas9 system is not a silence editing tool, so if a wild type phenotype was observed from 48 hours it´s a sign of mix population.
The MS didn’t have lines during the main text making it difficult to establish the minor errors and adjustment required. Please, find the reviewed considerations in notes highlighted in the attached revision version of the MS.

Author Response
Dear Reviewer: Thank you very much for your comments and the corrections mede on the text of our manuscript.
Reviewer 2
- All authors are from Cinvestav? Please, give the full address of each department.
R= I confirm that all the authors of this article belong to CINVESTAV-IPN, with the address listed at the end of the department.
- Change "to know" in the first line of the abstract to "to better know".
R= Recommendation followed
- Please, add the number of the lines in thu full MS to failitate the revision.
R= Recommendation followed
- "Their" din´t fit well in the begin of the third phrase of the abstract. Consider change to "These genes" or "The study of this locus"...
R=Recommendation followed
- " in vivo" also in italic, such as "in vitro"
Recommendation followed
- Keywords should be not present in then title to help researchers to find your MS. Please, consider change for alternative words
R= We have changed some of the keywords, however, we consider that the rest of them are necessary to find the main issue of the manuscript.
- Leishmania is inside Trypanosomatidae family, which is not in italic. Consider just genera names in the last paragraph of the introduction or remove "Leishmania spp." from the sentence.
R= We consider it’s necessary to mention Leishmania, even when it belongs to the Trypanosomatide family, because CRISPR-Cas9 has been successfully used in this particular parasite.
- ehcp112 is not introduced until the last sentences of the last paragraph of the Introduction section. Please, hidhlight the importance of this virulent gene.
R= We added a sentence highlighting the importance of ehcp112 gene. Recommendation followed
- Please adjust the website font to the same of the entire text
R= Recommendation followed
grna.ctegd.uga.edu
- What "score" means in the Table 1?
R= Score: A score closer to 1 presents a higher probability of hybridization with DNA target according to the software used.
We added this note at the bottom of table 1.
- "0 uM"? Is it right? Add the correct concentration.
R= Thank you for this observation we corrected it to 10uM
- Suppementary table? The "STable" was not fine.
R= We had an error, it should be only Table 2, which has already been corrected.
- Is it the "STable 2"? Correct to just table 2 in the main text and in the title of the table, removing "Scheme".
R= Recommendation followed, change de name for Table2.
- Please, establish the number of colonies used to inoculate in LB medium.
R= Recommendation followed: We used only one colony for experiment to inoculate 1L of LB medium
- "RT" is room temperature? If so, provide the full name at the first time it appears.
R= Recommendation followed. Room temperature (RT)
- What is "RT" in "RT-qPCR"? Room temperature? I believe it´s reverse transcriptase. Please, decide how "RT" abbreviation will be used.
R= We added the full name of RT-qPCR (reverse transcription quantitative polymerase chain reaction)
- The housekeeping gene is neither provide in the "STable 2" nor referenced in the methodology. Please provide the sequence or add the reference of the oligonucleotide used to amplified ehs40s.
R= Recommendation followed in the table 2.
- Room temperature, real time or reverse transcriptase? Please, provide what is "RT" in entire main text.
R= This suggestion has already been done. The definition RT-qPCR (reverse transcription quantitative polymerase chain reaction) has been added to the text where it first appears.
- The entire first paragraph of the Results section is not a result and should be not stated in this section.
R= This suggestion has been already followed and the first paragraph of Results have been eliminated.
- Please, provide the description of what is the red letters in the last sequence of the Figure 1B.
R= Red letters correspond to DNA templates containing sequences of the T7 promoter as described in the figure legend 1B.
- PCR reactions should have negative controls which are not provide in all the MS results!! Adjust all PCR figures
R= Suggestion considered, the negative control (ctl-) was added in figure 2D and MS y results
- Figure 3 is well done, but don´t add any relevant information about the study. Please, remove it.
R= Thank you for your comment. However, we believe that this figure is didactic and important for people that is not familiar with the CRISPR-Ca9 strategy. For this reason, we ask to this reviewer that accept our figure because it shows how RNPs were introduced into the medium using the modified technique for CRISPR in amoeba, that differs from other methodologies that use plasmids to introduce the RNPs.
- Change the statement "Our results altogether" to "Altogether, our results..." or just "Our results"...
R= Suggestion considered
- The images didn´t show exactly what is stated in the description of the results. The magnified cell was chosen and not necessary illustrate the others in the complete image. Consider remove this subsection and just mention the results as data not shown.
R= We have removed the magnified pictures from Figure 6.
R= Recommendation followed
- Add the negative control the reaction in the gel.
R= Negative control was added in Fig. 7A
- If the wild type phenotype of EhCP112 was recovered the knockout didn´t work. CRISPR/Cas9 is not a silence methodology, and this is the main issue of this MS.
R= Thanks for all your excellent comment. Yes, you are right, however, in our work we obtained transitory knocking down of the gene that was reflected in the non-expression of the protein. This allows to make studies on the protein function during this lapse. Currently, we are working using cloned populations, however, cloning in semisolid agar takes long time and the clones require to be characterized before using them for virulence-related studies. In addition, we do not discard the homologous recombination in the parasite that, according to many authors present specific characteristics, some of them not completely studied. Our studies using E. histolytica clone A, indicate that knock out should work better with a clone, but results are not complete yet for this paper. We include here a preliminary result using a cloned population for CRISPR-Cas9 experiments.

Reviewer 3 Report
Comments and Suggestions for Authors
This work provide the first demonstration of CRISPR-Cas9–mediated editing of an endogenous gene in Entamoeba histolytica, targeting the ehcp112 virulence gene. By using Cas9 ribonucleoprotein complexes delivered through a non-aggressive soaking method a transient knockout of ehcp112 was achieved, accompanied by phenotypic changes and altered expression of neighboring genes within the V1 locus. Although the effect is short-lived (approximately 24 h) and heterogeneous across the population, this study establishes a proof of principle that E. histolytica DNA can be modified using CRISPR-Cas9. These findings provide a foundation for the development of more stable and efficient gene-editing strategies in this parasite, which will be essential for dissecting virulence mechanisms and exploring therapeutic applications.
The knockout effect of ehcp112 lasted only ~24 hours, after which gene expression and protein production were restored. This raises concerns about the stability and reproducibility of the method. Can you clarify what strategies can be envisioned to obtain stable knockouts in E. histolytica (e.g., repeated RNP delivery, clonal selection, or integration of repair templates)?
The experiments were performed on bulk cultures, which are known to be genetically and phenotypically heterogeneous. This makes it difficult to determine the editing efficiency across individual cells. Have you considered single-cell deep sequencing approaches to measure the true editing frequency and reduce variability?
While sgRNA design software was used to minimize off-target sites, the study did not provide experimental validation of potential off-target cleavage. Did you assess possible off-target effects in the genome, and discuss how might this can be addressed this in future studies (e.g., targeted resequencing of predicted sites)?
The knockout of EhCP112 increased total protease activity in trophozoite extracts, suggesting compensatory mechanisms. However, the specific proteases responsible were not identified. A proteomic or transcriptomic profiling would be useful to determine which proteases are upregulated in the absence of EhCP112?
Editing of ehcp112 also reduced the expression of neighboring genes (ehadh and ehrabb). While this may reflect promoter overlap, alternative explanations such as indirect regulatory effects cannot be excluded.
Author Response
Reviewer 3
The major issue constitutes the efficacy of your ehcp112 knockout, since the absence of the protein was just validated 24 hours after CRISPR/Cas9 edition. The addition of a resistance marker, or parasites. CRISPR/Cas9 system is not a silence editing tool, so if a wild type phenotype was observed from 48 hours it´s a sign of mix population.
Dear reviewer: Thank you very much for your comments. Absolutely, we agree with you that to have a drug-resistance marker, would be great for these experiments. However, as we did not use plasmid or another vector for RNPs introduction, the methodology for adding a drug resistance marker has not been implemented nor designed. We have produced emetine-resistant mutants using EMS as mutagen. But given that the EMS generates pleiotropic genetic changes, it is not suitable for our experiments. Currently, we are working in the design of the best pathway to do this. Since, this is the first time CRISPR/Cas9 gene editing system has been implemented in E. histolytica, there are still aspects that need to be fine-tuned to achieve a stable knockout. This time, we used the HM1:IMSS strain, and the RNPs were incorporated from the medium, and we believe that due to this methodology was possible to avoid the damage produce to the cells by the Cas-9 enzyme, described for other systems. The gene edition can be repaired by homologous recombination, and this parasite has an efficient DNA repair machinery. To address this, we used the clone A population. We achieved mutants for a longer time. However, this is an old clone and might have gained changes in the DNA during all this time. Even though, the effect lasted at least 72h. We are working on this issue with the aim of developing a stable mutant. We attach the Wb image, which shows the absence of the EhCp112 protein in the CRISPR-treated clone A.

These findings provide a foundation for the development of more stable and efficient gene-editing strategies in this parasite, which will be essential for dissecting virulence mechanisms and exploring therapeutic applications.
We coincide with your appreciation that CRISPR-Cas-9 strategy implementation in E. histolytica will allow to make further studies on virulence factors in this parasite..
Can you clarify what strategies can be envisioned to obtain stable knockouts in E. histolytica (e.g., repeated RNP delivery, clonal selection, or integration of repair templates)?
Yes, we need to obtain clonal populations and to develop drug-resistant clones that can be used in future experiments.
Have you considered single-cell deep sequencing approaches to measure the true editing frequency and reduce variability?
It is an excellent idea, but for the moment we have not done that.
While sgRNA design software was used to minimize off-target sites, the study did not provide experimental validation of potential off-target cleavage. Did you assess possible off-target effects in the genome, and discuss how might this can be addressed this in future studies (e.g., targeted resequencing of predicted sites)?
The in vitro assays before the in vivo ones, gave us better chances for minimizing the off-target sites. In fact, we probed 10 sgRNAs and only two efficiently edited the DNA.
A proteomic or transcriptomic profiling would be useful to determine which proteases are upregulated in the absence of EhCP112?
Yes, you are right, transcriptomic profiles will be useful to determine which proteases are upregulated in the absence of EhCP112. This will be the focus of future experiments in subsequent studies.
Alternative explanations such as indirect regulatory effects cannot be excluded.
In fact, we cannot rule out the existence of other indirect regulatory effects.
Round 2
Reviewer 2 Report
Comments and Suggestions for Authors
Dear authors,
Congratulations for the study one more time. I'm not entirely convinced about your conclusions with a transitory knockout strain, and also believe you should correct minor issues not addressed: keywords, Leishmania in the introduction and Figure 3. However, considering your efforts to improve the quality of the MS, I will recommend the approval of the study for publication.